# Statistical Physics of Evolving Systems

**DOI:** 10.3390/e23121590

**Published:** 2021-11-27

**Authors:** Arto Annila

**Affiliations:** Department of Physics, University of Helsinki, 00014 Helsinki, Finland; arto.annila@helsinki.fi; Tel.: +358-44-204-7324

**Keywords:** dissipative systems, evolution, free energy, natural selection, power laws, quantum of action

## Abstract

Evolution is customarily perceived as a biological process. However, when formulated in terms of physics, evolution is understood to entail everything. Based on the axiom of everything comprising quanta of actions (e.g., quanta of light), statistical physics describes any system evolving toward thermodynamic balance with its surroundings systems. Fluxes of quanta naturally select those processes leveling out differences in energy as soon as possible. This least-time maxim results in ubiquitous patterns (i.e., power laws, approximating sigmoidal cumulative curves of skewed distributions, oscillations, and even the regularity of chaos). While the equation of evolution can be written exactly, it cannot be solved exactly. Variables are inseparable since motions consume driving forces that affect motions (and so on). Thus, evolution is inherently a non-deterministic process. Yet, the future is not all arbitrary but teleological, the final cause being the least-time free energy consumption itself. Eventually, trajectories are computable when the system has evolved into a state of balance where free energy is used up altogether.

## 1. Introduction

Impressed by Darwin’s tenet, Boltzmann searched for physical laws underlying evolution. However, as an atomist, he did not make any profound difference between animate and inanimate but reasoned evolution of every kind following the same law. As the renowned passage reveals, Boltzmann recognized entropy and free energy as the concepts whose changes characterize evolution, “The general struggle for existence of living beings is therefore not a fight for energy, which is plentiful in the form of heat, unfortunately untransformably, in every body. Rather, it is a struggle for entropy that becomes available through the flow of energy from the hot sun to the cold earth” [1].

Despite Boltzmann’s insight [2,3], the relationship between increasing entropy and decreasing free energy has remained obscure [4]. Early on, Carnot understood that a change in entropy, *dS* = δ*Q*/*T*, follows from a change in energy δ*Q* per temperature *T* (i.e., average energy when multiplied with Boltzmann constant *k_B_*). Yet, in the quest for the microscopic formulation, Clausius, Helmholtz, and Boltzmann equated entropy, *S* = *k_B_*ln*W*, with disorder enumerating the so-called microstates, *W* (i.e., energetically indistinguishable constituent configurations) [5]. Moreover, Schrödinger stating that life feeds on negative entropy led to a worsened confusion [6].

Boltzmann posited statistical mechanics to be the theory underlying thermodynamics but acknowledged failing in formulating it [7]. His equation for entropy is a condition of thermal equilibrium rather than an equation of state [8,9,10], and hence is limited to closed and isolated systems. Even so, it paved the way to modern near-equilibrium methods [11,12,13]. So, these models, too, fall short in describing the grand changes of open systems. In essence, substantial fluxes of matter and radiation, typical of growing organisms and their populations, have remained unformulated in theoretical terms until our time. Thus, instead of being a mere mathematical model of data [14], a theory of evolution ought to be placed on an axiom to be falsifiable with the data it aims to explain [15,16].

Such a comprehensive theory is not inconceivable. On the contrary, it seems even indispensable since data is surprisingly similar across scales and scopes. As noted already by Galilei [17], Snell [18], Thompson [19], and Huxley [20], and in our time vindicated by complex system science [21], data of various kinds closely follow power laws. In other words, charts without legends and labels give us no clue whether data are about molecules, genes, organisms, or populations [22]. This isometric scaling, also known as allometry [23,24], results from a chain of events where each stage of development follows from all the previous steps (i.e., evolution).

The universality of patterns, including sigmoidal curves, skewed distributions, logarithmic spirals, and even regularity of chaos [22,25,26], suggests that animate and inanimate are indeed basically one and the same. Thus, the evolution of a species on an island should not be theorized to be any different (e.g., from the cooling of tea in a cup) [27]. As Boltzmann surmised, difference, be it in temperature, chemical energy, or another form of energy, diminishes with time (in fact, in the least time). Therefore, all kinds of systems evolve similarly. They all are on their way to attaining thermodynamic balance with their surrounding systems in the least time.

In setting up statistical physics of open, evolving systems of any kind, let us employ the old atomistic axiom, also favored by Boltzmann. Of course, by today, it is known that atoms comprise constituents, but the ultimate constituent is not generally known.

Considering that evolving systems absorb and emit photons in changes of state, let us assume, as Newton implied [28], that everything comprises photons. Despite the logical consistency, the all-inclusive axiom, reworded by Lewis [29], may seem unwarranted since the number of photons is not conserved in calculations employing quantum mechanics creation and annihilation operators. While effective theories serve for calculating, calculations themselves do not refute Lewis’ postulate. The number of light quanta does not have to be non-conserved even though photons appear out of the vacuum and disappear into it. Namely, it all depends on what the vacuum is. The photons could appear out from the vacuum comprising photons in pairs with opposite phases, or the photons could disappear into it by pairing anew [30,31]. The inference agrees, for example, with the dynamic Casimir effect, where visible photons emerge out of the vacuum two at a time [32]. Also, the degeneracy of two for each energy level in the Bose-Einstein statistics implies a paired-photon substance. Since there is no compelling experimental evidence against the photon being the fundamental constituent, let us proceed from this axiom to derive the state equation.

## 2. Materials and Methods

In a complete theory, every element of reality has a counterpart [33]. Accordingly, any system can be described by the same theory, assuming that the fundamental elements (i.e., the quanta) embody everything. The atomistic axiom implies that one and the same state equation accounts for all systems, and its derivative, the equation of motion, accounts for the evolution of any system.

### 2.1. The State Equation

The scale-free state equation can be deduced from the general diagram of energy levels of a system (Figure 1) in terms of probability, *P*, the central concept of statistical mechanics [3,34,35]. The probability of any one entity, indexed with *j*, _1_*P_j_* = *ϕ*_1_*ϕ*_2_*ϕ*_3_… = ∏*_k_**ϕ**_k_*, as a product, ∏*_k_*, of substrates, labeled with *k*, guarantees that if a *k*-substrate is absent entirely, i.e., *ϕ**_k_* = 0, also _1_*P_j_* = 0. For example, a given protein in a cellular system could not be present if any of its constituents, such as an essential amino acid, was absent altogether.

The success of statical mechanics stems from the comprehensiveness of the underlying axiom. Irrespective of the complexity, it is not necessary to explicitly know the components, *k*, in the product, ∏*_k_*, to count them all since all entities ultimately comprise the same basic constituents.

When the system holds interchangeable entities in numbers, *N_j_*, say, a cellular system holds many copies of a protein, the population probability, *P_j_* = [_1_*P_j_*][_1_*P_j_*][_1_*P_j_*]…/*N_j_*! = [_1_*P_j_*]*^Nj^*/*N_j_*! is _1_*P_j_* in power *N_j_*. Once more, the product guarantees that if any one entity is absent entirely, i.e., _1_*P_j_* = 0, also *P_j_* = 0. Since the order among identical entities makes no difference, indistinguishable configurations (i.e., microstates) should not be counted as different states unless truly independent [36]. Hence, *P_j_* is scaled by the permutations, *N_j_*!

To conclude, the total probability, *P*, of the system is the product, ∏*_j_*, over *P_j_*
(1)$$P=\underset{j}{\prod }P_{j}=\underset{j}{\prod }{\left[\underset{k}{\prod }\phi_{k}\right]}^{N_{j}}/N_{j}!\;,$$
where the factor, *ϕ**_k_* = *N_k_*exp[(−Δ*G_jk_* + *i*Δ*Q_jk_*)/*k_B_T*], encompasses the substrates, *N_k_*, each of free energy, −Δ*G_jk_* + *i*Δ*Q_jk_*, per, *k_B_T*, the average energy. When fluxes of quanta perturb *k_B_T* only a little, the system changes gradually, as if in a continuous manner. Then, an energy difference can be approximated with the selfsimilar exponential function *de^x^*/*dt* = *e^x^* [37]. The energy difference, Δ*G_jk_*, between the substrate, labeled with *k*, and the product, labeled with *j*, is spanned by Δ*Q_jk_*, such as insolation, from the surroundings. The quotient, *i*, distinguishing vector potential from the scalar one, indicates that the system is open for the flows of quanta that couple to a *jk*-transformation. For example, insolation makes photosynthesis happen and dissipation enables metabolism.

The equation of state (Equation (1)) is the principal result of statistical physics of evolving systems. Its logarithm (ln) is the familiar additive, Σ, measure of a state, known as entropy, *S*, when multiplied by *k_B_*
(2)$${S=k}_{B}{\ln P=k}_{B}\underset{j}{\sum }{\ln P}_{j}\approx \frac{1}{T}\underset{jk}{\sum }N_{j}\left({-\Delta \mu }_{jk}{+i\Delta Q}_{jk}{+k}_{B}T\right)$$
where *μ_k_* = *k_B_T*ln[*N_k_*exp(*G_k_*/*k_B_T*)] is the potential bound in the *k* substrate, *μ_j_* is the potential bound in the *j* product, and their difference is Δ*μ_jk_* = *μ_j_* − *μ_k_*. The form of chemical potential qualifies for any bound potential, postulating everything comprising the same fundamental entities, the quanta. The surrounding quanta, carrying energy Δ*Q_jk_*, that couple to the *jk*-transformation contribute to the system’s energetic status. The approximation sign stands for Stirling’s approximation for factorial, ln*N_j_*! ≈ *N_j_* ln*N_j_* − *N_j_*, in Equation (2). It is excellent for a many-body system.

It is worth emphasizing that here entropy does not measure disorder. Instead, the system’s total energy, *TS*, includes the system-bound, Σ*N_j_k_B_T*, and free, Σ*N_j_*(−Δ*μ_jk_* + *i*Δ*Q_jk_*) energy. Thus, the system keeps changing until free energy is consumed altogether. Only then does the textbook form of entropy, *S* = *k_B_*Σ*N_j_*, apply.

### 2.2. The Evolutionary Equation

For a statistical system, the step-by-step changes in *N_j_* can be given to a good approximation by continuous differentials, *dN_j_*. Thus, the total energy of the system, *TS*, evolves with time, *t*,
(3)$$T\frac{dS}{dt}=T\underset{j}{\sum }\frac{dS}{{dN}_{j}}\frac{{dN}_{j}}{dt}=\underset{jk}{\sum }\frac{{dN}_{j}}{dt}\left({-\Delta \mu }_{jk}{+i\Delta Q}_{jk}\right)$$
when free energy, −Δ*μ_jk_* + *i*Δ*Q_jk_*, is consumed in transformations from *N_k_* to *N_j_* or vice versa. The average energy, *k_B_T*, is not explicitly differentiated using the chain rule since its variation contains variations in *dS*.

The change in a population, *N_j_*,
(4)$$\frac{{dN}_{j}}{dt}=\frac{1}{k_{B}T}\sum_{k}\sigma_{jk}\left({-\Delta \mu }_{jk}{+i\Delta Q}_{jk}\right)$$
is proportional to free energy by mechanistic factors, *σ_jk_* > 0 [12]. A factor, *σ_jk_*, such as a catalyst, facilitates the *jk*-transformation from *N_k_* to *N_j_*, and the other way around. The natural process naturally selects the most efficient mechanisms of free energy consumption to attain balance in the least time. In a sense, the various processes by Equation (4) compete for the flows of quanta. Thus, the least-time principle can be recognized as the long-yearned teleological imperative of evolution, the final cause [38]. The quest for attaining balance in the least time explains, for example, the proliferation of insolation-absorbing compounds during abiogenesis [39,40].

From the thermodynamic perspective, life is a response of matter on earth to sunlight [41]. Over the eons, the bound form of energy has increased as organic material has been deposited in the curst. Conversely, interfering with the global quest for balance by consuming fossil fuels and disrupting energy transduction mechanisms by deforestation [42] comes with catastrophic consequences.

As can be proved by squaring the free energy terms in Equations (3) and (4), entropy cannot decrease, *dS* ≥ 0, not even momentarily or at the cost of an increase elsewhere. Note, when squaring, that the *jk*-basis is orthogonal. This reasoning about *dS* ≥ 0 follows from the conservation of quanta. The quantized fluxes level out energy differences between the system and its surroundings. The inference is supported by empirical evidence. Both living and non-living systems display the same patterns [22,25,26].

The courses of evolution, development, differentiation (i.e., transformations of any kind) cannot be computed, for variables in Equations (3) and (4) cannot be separated. Yet, simulations according to Equation (4) demonstrate that the least-time processes produce standards, such as chirality consensus, nearly lognormal distributions, sigmoidal growth curves, allometric scaling, oscillations, and regularity of chaos [27,42,43,44].

### 2.3. The Continuous Equation of Motion

Many quantized processes seem almost like continuous motions. Thus, expressing the scalar potential *μ_j_* = (∂*U*/∂*N_j_*) and radiation *Q_j_* = (∂*Q*/∂*N_j_*) in terms of continuous potentials *U* and *Q* serves to cast Equation (3) to
(5)$$T\frac{dS}{dt}=\sum_{jk}\frac{{dN}_{j}}{dt}\left(-\frac{\partial U}{\partial N_{j}}+i\frac{\partial Q}{\partial N_{j}}\right)=-\frac{\partial U}{\partial t}+i\frac{\partial Q}{\partial t}=\frac{d}{dt}2K$$
as *dN_j_* depends on ∂/∂*N_j_* while being independent of ∂/∂*N_k_*. Since fluxes of photons of *Q* transform *U* and *K*, *TdS* corresponds to the change in kinetic energy *d*2*K*. In other words, energy and period, as well as momentum and wavelength, change when the quanta settle on new trajectories. For example, when an electron settles on a higher orbit, the Coulomb field in an atom changes.

The integral form of the continuous equation of motion (Equation (5)) leads to Maupertuis’ principle of least action by requiring vanishing variation of the integrand of ∫**p**·*d***x** = ∫2*Kdt* that totals quanta with momenta, **p**, on paths, **x**, or equivalently with kinetic energy, 2*K*, on periods, *t* [45]. The free-energy variational principle describes complex processes [46], such as morphogenesis [47]. Moreover, the revived interest in the Maupertuis action has led to recognizing causality as its partial derivative of along the spatial coordinate with the smallest momentum change along the trajectory [48] and connection between time and thermodynamics for equilibrium phenomena and its extension to irreversible processes assuming local equilibrium [49].

In contrast to the closed, stationary Lagrangian form, the original Maupertuis form is open for evolution. Mathematically speaking, the dissipation, ∂*Q*/∂*t* ≠ 0, moves integration limits while integrating since forces change motions, changing the forces, and so on. The inseparable variables make it impossible to solve the evolutionary equation. Thus, the future remains non-determined. However, the least-time maxim distinguishes evolution from an (illusory) indeterminate (i.e., random) motion.

The equation of motion (Equation (5)) can also be derived from Newton’s second law
(6)$$F=\frac{d}{dt}p=ma+v\frac{dm}{dt}|\;v\;v\cdot F=v\cdot \frac{d}{dt}p=\frac{dx}{dt}\cdot ma+v\cdot v\frac{dm}{dt}=-\frac{dU}{dt}+i\frac{v^{2}}{c^{2}}\frac{dE}{dt}=-\frac{dU}{dt}+i\frac{dQ}{dt}$$
by multiplying the change, *d***p**/*dt*, in momentum, **p** = *m***v**, with velocity, **v**, and decomposing kinetic energy, 2*K* = **v ∙ p** = ∑*v_j_mv_k_*, that vanishes for *j* ≠ *k*, and taking into account that *d***v**/*dt*
**∙ p** = 0 since acceleration **a** = *d***v**/*dt* ⊥ **v**. The change in mass, *dm*/*dt* = *dE*/*c*^2^*dt* = *dQ*/*v*^2^*dt,* denotes, in geometric terms, changes in the quantized trajectories as they open up and dissipate quanta into the surroundings. Concurrently to any state transformation (e.g., nuclear, chemical reaction, also mass changes, *dm*). Accordingly, masses change until the system develops into a steady state. Mass-energy equivalence, *E* = *mc*^2^, the relativistic formula, is understood here to follow from the action, *Et* = *mc*^2^*t* = *px*. Thus, the 2nd law of thermodynamics, the original (Maupertuis’) principle of least action, and the complete form of Newton’s second law of motion are the same law.

Eventually, the system reaches balance in its surroundings. As the net flows between the system and its surroundings have died out, the evolutionary equation (Equation (6)) reduces to the integrable virial theorem, 2*K* + *U* = 0. The steady-state system totaling *n* quanta with kinetic energy, 2*K*, says by Noether’s theorem, 2*Kt* = *nh*, that the quanta orbit within characteristic periods, *t*. Time invariance corresponds to constant energy. Symmetry remains unbroken.

## 3. Results

Since statistical physics of open evolving systems is based on an axiom, data in its entirety is free to verify and falsify this many-body theory. As it turns out, data of statistical systems follow closely but not precisely ubiquitous power laws. Most notably, data deviate both at low and high ends. This implies that the power law is an approximation of a sigmoid curve that accumulates from a skewed, nearly lognormal distribution [50]. These characteristics can be proven to result from Equations (3)–(6) through mathematical analysis [51].

### 3.1. The Ubiquitous Patterns

The characteristic S-shape of a growth curve can be seen from Equation (4). In the beginning, there is a wealth of resources, i.e., free energy for growth. So, we may assume that mechanisms, Σ*_k_**σ**_jk_*, limit free energy, −Δ*μ**_jk_* + *i*Δ*Q_jk_*, consumption, and define
(7)$$\frac{d}{dt}\frac{1}{k_{B}T}\sum_{k}\left({-\Delta \mu }_{jk}{+i\Delta Q}_{jk}\right)=\frac{{dN}_{j}}{dt}\frac{d}{{dN}_{j}}\frac{1}{k_{B}T}\sum_{k}\left({-\Delta \mu }_{jk}{+i\Delta Q}_{jk}\right)\approx \sum_{k}\sigma_{jk}\;\Rightarrow \frac{{dN}_{j}}{dt}=\sum_{k}\sigma_{jk}N_{j},$$
using *d**μ**_j_*/*dN_j_* = *d*(*G_j_ + k_B_T* ln*N_j_*)/*dN_j_* = *k_B_T*/*N_j_*, since *μ**_k_*, *Q**_j_* and *Q**_k_* have no explicit but only a stoichiometric dependence on *N_j_*. The initial growth by Equation (7) is approximately exponential since initially, the amount of free energy seems as if infinite. In turn, when the free energy dwindles down, the final growth decreases slowly, almost exponentially.

The growth between the initial and final phase trends a power law. Expressing *N_j_* as the product of the basic elements, *N*_1_, and using the atomistic axiom, *N_j_* = ∏*_k_**ϕ**_k_* = *α**_j_**N*_1_*^j^*, where *α**_j_* = ∏*_mn_*exp[Σ*N_j_*(−Δ*μ**_mn_* + *i*Δ*Q_mn_*)/*k_B_T*] contains the free energy terms that force the assembly of *N_j_* from *N*_1_ through various *mn*-transformations, the change,
(8)$$\frac{{dN}_{j}}{dt}=j\alpha_{j}N_{1}^{j-1}\frac{{dN}_{1}}{dt}=j\frac{N_{j}}{N_{1}}\frac{{dN}_{1}}{dt}\;\Rightarrow \;\frac{{dN}_{j}}{N_{j}}=j\frac{{dN}_{1}}{N_{1}},$$
when integrated, follows a power law ln*N_j_* = *j* ln*N*_1_ + constant. The form is familiar from various self-organizing processes, including technological ones such as the one leading to Moore’s law [52]. The power law in the continuous form, *d* ln**p** = *d* ln**v** + *d* ln*m*, follows from dividing Newton’s second law of motion by momentum, **p**.

When the assumption of a nearly constant change in free energy does not hold, we may model the change by adding the term ^−^*β**N_j_* to Equation (7)
(9)$$\frac{{dN}_{j}}{N_{j}}\approx \left(\underset{k}{\sum }\sigma_{jk}-{\beta N}_{j}\right)\Rightarrow N_{j}\left(t\right){=N}_{j}\left(t_{o}\right)\left(\underset{k}{\sum }\sigma_{jk}-{\beta N}_{j}\left(t_{o}\right)\right),$$
where the population, *N_j_*(*t*_o_), at an instant, *t*_o_, determines the population, *N_j_*(*t*), at a later instant, *t*. By this model [53], evolution is almost predictable when the change in free energy compared with average energy, |(−Δ*μ**_jk_* + *i*Δ*Q_jk_*)/*k_B_T*| << 1, is small [51]. And when not, oscillations, bifurcations, and chaos occur, exemplified by soaring and sinking populations.

Logarithmic, exponential, and truncated distributions and their power-law-like cumulative distribution functions are mathematical models of the physical processes given by Equations (3) and (4). The renowned deterministic models mostly tally data, but they do not explain phenomena since, in reality, the chain of events is fundamentally unpredictable. However, non-determinism does not follow from the complexity of a system or ambiguity in its initial conditions but from mutual dependencies. Equations (3)–(6) cannot be solved, except at balance, since the variables cannot be separated. Everything depends on everything else. The cherished assumption ceteris paribus does not hold.

When the system evolves gradually, the change in energy is small compared with the average energy, |(−Δ*μ**_jk_* + *i*Δ*Q_jk_*)/*k_B_T*| << 1. As the variation, *n*, is small, *n* << *j*, around a representative, mean, or an average factor, *ϕ_j_*, the distribution of factors, given in logarithmic terms, ln*ϕ_j_* = *j* ln*ϕ*_1_, of the elemental factor, *ϕ*_1_,
(10)$$\ln\phi_{j-n\ldots j+n}=\ln\phi_{j}+\underset{n}{\sum }\ln\phi_{1}\;,$$
is approximately logarithmic [54,55]. The typical form, *j*, can be recognized in each member within the distribution *j* ± *n*. For example, all sized birds look like a bird and not fish. Accordingly, if the weights of birds and fish were plotted on the same graph, we would see two distributions, namely one about a typical bird and the other about an average fish.

Spirals, such as shells, flower beds, whirls, cyclones, and galaxies, are also approximately lognormal distributions in polar coordinates (i.e., energetically optimal shapes) [51,56].

### 3.2. Non-Deterministic Motion

The non-deterministic nature of evolution is apparent from Equation (4). The change, *dN_j_*, cannot be separated from its driving force since Δ*μ_jk_* is a function of *N_j_*. As Equation (3) cannot be solved, the course of events is, so to say, non-deterministic, non-computable, and intractable [57]. This conclusion contradicts the common misconception that evolution would be a random (i.e., an indeterministic process). A mutation in a gene may appear such as an arbitrary event. However, it also has its cause, such as natural background radiation or exposure to a chemical agent. The significance of a mutation depends on the free energy it may unleash when the gene is expressed. Thus, it is the consumption of free energy, not the facilitating mutations, that drives evolution. In short, evolution follows forces.

While evolution is teleological in the least-time sense, the interdependency among forces and changes in motion excludes evolution targeting preset goals. Instead, at every step, flows of quanta choose the paths for the least-time free energy consumption. Eventually, a thermodynamic balance will be attained, but the state cannot be predicted from the initial condition. Moreover, the balance may be only temporary. When new forms of free energy appear or new means to consume free energy emerge, the system evolves again. For example, the invasion of a new species and the invention of technology cause changes.

Evolution as a non-deterministic thermodynamic process also makes it clear why hierarchical structures reappear after catastrophes, not exactly identical but closely similar to the demolished ones [27]. Moreover, as everything depends on everything else, the more versatile thermodynamic systems are, the more resilient they are to changes. Conversely, systems poor in diversity, say, arctic ecosystems and small economic systems, are susceptible to major changes.

## 4. Discussion

Statistical physics of open systems concurs with the tradition of theorizing and contrasts with the trend of modeling. In ancient Greek, theōría meant viewing, considering, examining from a viewpoint. From the axiomatic perspective, it only remains to check whether observations match the projection. If not, the theory is false. However, unlike an effective theory (i.e., a mathematical model), the axiomatic theory cannot be fixed and fitted to the data. Instead, a new axiom must be put in place in hopes of a better theory making sense of what is seen. Consequently, only comprehensive axioms, such as the atomistic axiom underlying a complete theory, are of interest and impact (e.g., as we struggle to transform our way of life for sustainable subsistence) [58].

Statistical physics of open systems deserves benchmarking against Kuhn’s five criteria of a sound scientific theory [59]. First, the theory must be accurate within its domain. The results should correspond to what is observed and measured. Clearly, the atomistic axiom defines the domain of everything that there is. Accordingly, it may take a considerable effort to examine nature from its tiniest elements to its enormous extents before finding a case that proves the theory false.

Second, the theory must be consistent not only in itself but in relation to the prevailing theories. As shown above, statistical physics of open systems restates the complete Newton’s second law and the original principle of least action. However, the correspondence with modern physics is an issue. Yet, the comparison between the axiomatic theory of statistical physics and the effective theory of the standard model of particles is not straightforward. Unlike the traditional theory, the instrumental tenet aims to match rather than explain data. The standard model of cosmology is also an effective theory as it is parametrized with dark matter and dark energy to match with observations. However, the axiom general relativity that the laws of physics are the same for all observers, regardless of their motion, is consistent with the atomistic axiom of statistical physics.

Third, the theory should have a broad scope. It should explain much more than its first objective. Here the first objective was to derive the theory of evolution, customarily understood as a theory of biology, from physics. However, the scope of the theory expanded beyond the first objective as it was then realized that statistical physics of open systems by its universal axiom accounts for the evolution of any system.

Fourth, the theory should be simple and bring order to phenomena that otherwise would be individually isolated and, as a set, confused. Indeed, the statistical physics of open systems puts biological processes into the unity of all processes and brings order among all kinds of processes.

Fifth, the theory must be fruitful in revealing new phenomena and unknown connections among those already known. Indeed, it is eye-opening that the ubiquitous patterns (i.e., skewed distributions, sigmoidal cumulative curves, oscillations, and even chaos [60]) are manifestations of the universal imperative to consume free energy in the least time.

While this superficial inspection at hand seems approving and promising, it is worth recalling that Kuhn deemed it demanding to evaluate a theory’s accuracy, consistency, scope, simplicity, and fruitfulness. For instance, the sun-centered model was at odds with what was once thought about why stones fall, how water pumps operate, and how clouds move. So, let us recall that in the foreword of Novum Organum, Bacon urged to be open-minded for unconventional reasoning since conventions only appear as incontestable truths [61].

## Figures and Tables

**Figure 1 entropy-23-01590-f001:**
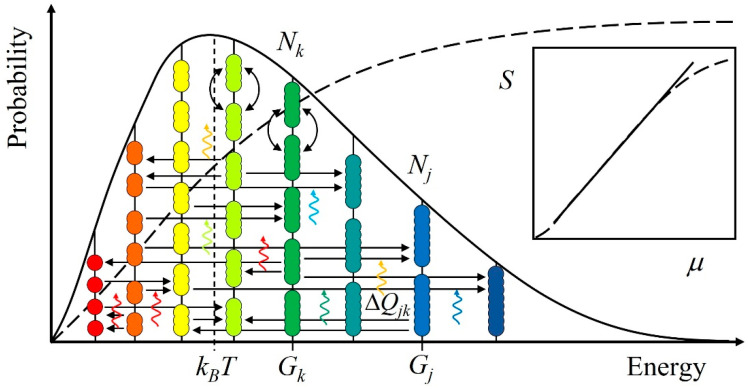
Any system can be depicted by the same general energy level diagram, assuming everything comprising quanta. Entities, in numbers, *N_k_*, with energy, *G_k_*, are on the same level. Their mutual exchange (bow arrows) brings about no change, so the average energy, *k_B_T*, does not change either. However, when quanta from surroundings with energy, Δ*Q_jk_*, (wavy arrows) couple to transformations from starting materials, *N_k_*, into products, *N_j_*, the entities move (horizontal arrows) from one level to another. Through transformations, the system and its surroundings move toward mutual thermodynamic balance. The system evolves toward higher average energy when the surroundings are higher in energy and vice versa. The logarithm of the sigmoid cumulative probability, *P*, distribution curve (dashed line) is entropy, *S* = *k_B_*ln*P*. On the logarithm-logarithm scale (inset), *S* vs. (chemical) potential energy, *μ*, closely follows a power law (i.e., a straight line).

## Data Availability

Not Applicable.

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
