# Peer review of "Statistical Physics of Evolving Systems"

_entropy, 2021, doi:10.3390/e23121590_

Round 1

Reviewer 1 Report

The paper is very interesting as usual for the Annila papers. Always prof. Annila suggests new ideas and comprhension of the reality.

I can only suggest some improvements:

  • Each physical quantity must be well defined after the equations and I suggest to introduce the unit of measurements
  • In Figure 1 I suggest to introduce the name of the x-axis
  • For the exchange quantities (heat and work), it is better to use \delta and not d, as path differentia

I thank Arto Annila to have submitted this paper to Entropy and the Editor to have involved me to review the paper.

I suggest to accept it for publication.

Author Response

Thank you very much for pointing out the shortcomings of the manuscript. I have paid particular attention to defining variables and figure axes. Moreover, I have tried to be consistent in my usage of continuous differentials as approximations of quantized motion and differences (delta). Furthermore, I realized that I should avoid using thermodynamic terms, like work, with an established meaning, but without dissipation, that might confuse readers. Also, heat with the standard meaning of motion might cause confusion as I have kinetic energy to motion. 

Yours sincerely,

Arto Annila

Reviewer 2 Report

The paper is interesting, but a bit densely packed. However, for someone familiar with the author's contributions in the past this shouldn't be a big issue. The paper argues that the least action principle is the primary idea that dictates all natural processes in this universe, including evolution and the emergence of hierarchical structures and self-similar complex patterns. I was a bit concerned as the action principle is a deterministic principle, purely mechanistic, but evolution is indeterministic. However, the author has tried addressing those questions as well in section 3.2. In connection to section 2.3, specially in the discussion of Maupertuis principle I would like to draw the author's attention to the recent study published in Entropy (Entropy 202123(9), 1212; https://doi.org/10.3390/e23091212) and Results in Physics (https://doi.org/10.1016/j.rinp.2020.103165). I think the connection between time, causality, entropy, and the Maupertuis principle is interesting specially in the context of evolution. Apart from this there are few minor grammatical errors, for example: the author uses the term 'selfsame' - I suggest self-similar; in the Discussion 'Forth' should be 'Fourth' in the fifth paragraph from the top. 

Author Response

I have revised the manuscript taking into the valuable corrections and comments. 

Specifically, I have expanded the manuscript a bit to improve on readability, clarity, and exactness of methods and results, corrected grammar and spelling. I have also added few more references to provide the readers with more context, in particular about the revived interest on the Maupertuis principle. 

Yours sincerely,

Arto Annila